# The Circular Economy: A Study on the Use of Airbnb for Sustainable Coastal Development in the Vietnam Mekong Delta

**Jianjia He** [1,2,*] and **Thi Hoai Thuong Mai** [1]

1    Business School, University of Shanghai for Science and Technology, Shanghai 200093, China;
     maithihoaithuong379@gmail.com
2    Supper Network Research Centre, University of Shanghai for Science and Technology,
     Shanghai 200093, China
*    Correspondence: hejianjia@usst.edu.cn; Tel.: +86-137-6458-1158

**Abstract:** The concept of the circular economy has become well known for its solution-oriented approach to transforming available resources into a closed-loop resource system. However, in the context of coastal areas, coastal resources seem to be ignored in the tourism production and consumption process. In relation to the 2030 Agenda for Sustainable Development Goals (SDGs), this article discusses how sharing economy practices may sustain coastal resources through ecotourism, applying a sharing-economy theory that emphasizes changes in the new form—a circular economy—rather than a single traditional Airbnb model or ecotourism model. This study proposes a coastal sustainable development structure model based on the integration between the sharing economy and ecotourism with three modes—positive economic effects, positive economic pressures, and sustainable coastal development—and uses coastal residents' expectations of their living conditions as moderating factors to investigate the impact of the circular economy on coastal sustainability. We developed a survey-based model that included 303 samples from the indigenous residents of 13 provinces throughout the Vietnam Mekong Delta. The results show that the integration of ecotourism with the Airbnb model has a positive effect on residents' living conditions, supporting sustainable local development. However, the advancement of technology and residents' awareness involves barriers to coastal development because the process of modernization is still limited in coastal areas. More specifically, in the case of the Vietnam Mekong Delta, our results suggest that limited technical knowledge and language ability stand as barriers to coastal businesses, showing that the lack of inter-regional connectivity limits the magnitude of local tourism in coastal areas. These findings are useful for assessing residents' living conditions so that coastal development can work towards poverty reduction. Finally, the establishment and expansion of policies by local authorities can be an indispensable part of coastal economic development by limiting the negative effects of the abuse of natural resources and facilitating family businesses in coastal zones in an effort towards the integration of economic development and social and environmental responsibility.

**Keywords:** circular economy; coastal ecosystem; ecotourism; Airbnb; sustainable coastal development

## 1. Introduction

Coastal resources are considered a crucial factor in the promotion of national coastal development, with a wide variety of benefits for human well-being. Coastal ecosystems are among the most productive on Earth, due to the variety of cultural diversity and rich resources [1,2] (estuaries and coastlines as well as adjacent land [3,4]). A coastal ecosystem is a major source of livelihood for coastal communities, providing a variety of attractive destinations and biological diversity that is considered a major factor to achieve the goal of ending global poverty, in all its forms, by 2030. More specifically, there have been an additional 88 to 115 million people falling into extreme poverty during the COVID-19

pandemic, with a standard of living of less than 1.90 USD a day [5]. Moreover, coastal zones do not seem to benefit from the intensive urbanization activities and technological breakthroughs of prosperous cities [6,7]. Therefore, the integration of responsible socioeconomic practices is considered in the process of effectively eliminating poverty [8,9]. Due to the importance of the Sustainable Development Goals (No poverty—SDG 1; Sustainable cities and communities—SDG 11; Responsible consumption and production—SDG 12), the circular economy investigates a social, cultural, and economic solution to the potential issues and opportunities in coastal areas [10,11]. The circular economy, in a business context, is seen as a solution-oriented approach to environmental sustainability, referring to production and consumption processes that limit the use of nonrenewable resources and produce almost no waste [12]. This paper plugs a research gap in that, in the context of coastal tourism, local sustainability may be promoted by the dominant features of the sharing economy.

This should be considered when assessing livelihoods in coastal areas, where most income fluctuations are closely linked to coastal ecosystems, as in the case of the fisheries of the Mekong River Delta, which have an estimated value of 9 billion USD per year [13]. Crop–aqua–agriculture in the Mekong Delta encompasses small-scale crop-based farming, pangasius catfish production, and fruit trees. More specifically, in 2018, the region accounted for 95% of the country's rice exports, as well as an annual seafood output of 3.5–4 million tons [14]. Aquaculture in the Mekong Delta accounts for 65% of the country's production, contributing 60% of Vietnam's fish exports, and fruit and vegetable exporters are responsible for 70% of the nation's fruit production [15]. The Mekong Delta tourism industry identified the development of river tourism, ecotourism, and garden tourism as attracting 47 million domestic and foreign visitors in 2019, an increase of 17.5% over 2018 [15]. However, due to the impact of the COVID-19 epidemic and social distancing regulations, in the first 4 months of 2020, the number of tourists to the Mekong Delta decreased significantly compared to 2019. Specifically, Kien Giang province, one of the key destinations for tourism in the Mekong Delta, attracted only 536,000 tourists, of which international visitors numbered 142,000, and domestic tourists numbered 394,000, decreasing by 54.7% compared to total visitors in the same period in 2019 [15]. The local tourism industry, too, continues to face significant challenges and limitations, necessitating the collaboration of all stakeholders involved in fostering local sustainable development, including policymakers, lodging providers, tourism agencies, and local residents [16,17].

In terms of market demand, homestay services continue to be a prominent option for local accommodation rather than traditional hotels, providing the best opportunity to engage with local communities and delve into the local culture. In 2019, Kien Giang, for example, had 241 active listings on Airbnb, and approximately 8.5–34.6% listing in other coastal cities in the Middle and South areas of Vietnam [18], demonstrating that Airbnb in the Mekong Delta has potential to be expanded.

In the private business sector, local-resource-based restructuring and diversification have been deemed important for regional economic development in the Mekong Delta by integrating fruit orchards, aquaculture, and animal farming, commonly known in Vietnamese as "VAC" farming [19]. VAC systems comprise three components: horticulture (gardening), aquaculture, and animal husbandry; therefore, they effectively use all the available land, air, water, and solar energy resources and also effectively recycle by-products and waste [20]. VAC layouts, and also the types of plants and animals in use, are highly varied between households and communities [21]. The components in the VAC system are closely related to each other and operated by the farmer, considered the most important element of the system, to regulate and support the survival of relationships in the VAC ecosystem [19]. More specifically, in the VAC system, fishponds use manure (discharged by livestock and poultry) as a source of food for fish in the pond. In addition, manure is used to fertilize the garden, ponds provide water for irrigation, silt soil adds nutrients for all plants in the garden, and, finally, surplus items from the vegetable garden are used to provide feed for livestock. Based on the VAC ecosystem, this study proposes a new model for the

circular tourist economy in the Mekong Delta, bridging the gap between sharing economics (represented by the Airbnb model) and tourism (represented by ecotourism) by effectively utilizing unused living space, fruit picking, fishing, cycling tours, and boating activities to create economic benefits that address the problem of poverty reduction in coastal zones (see Figure 1). It is important to note that ecotourism has a major environmental impact and can put great pressure on local resources [22]. Coastal resources are a core value of ecotourism, a subcomponent of sustainable tourism development based on the allure of its natural resources, which must be protected and preserved to support the transition to a more circular tourism economy in coastal zones [1]. More ecotourism activities are created, and the need to use coastal resources is expanded, resulting in increased water, energy, and food use, as well as noise and air pollution.

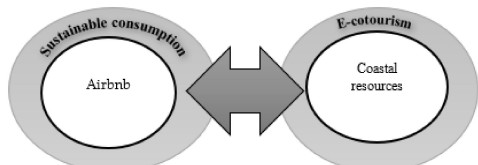

**Figure 1.** A VAC ecosystem-based proposal for the circular economy model.

As a result, this research focuses on incorporating the sharing economy into ecotourism activities in order to promote coastal development by enabling sustainable tourist development. Our emphasis is not on product innovation but on what tourists can accomplish. Therefore, the challenge is to understand how engaging travel experiences can arise from the circular tourism economy.

Previous researchers have mainly focused on analyzing the positive values of the sharing economy in urban and rural areas through both applied and empirical studies that could support the role of the economic dimension in improving living standards [23,24]. As the sharing economy is the guiding principle of a sustainable ecosystem, affecting local resources [25,26], it is important to study its role in coastal zones [27]. This is particularly promising because the values of the sharing economy are postulated as being important drivers of action in local sustainable development [28]. Contributing to this gap in the literature, this study focuses on the sharing economic theory and coastal ecosystems to examine whether this relationship positively benefits the livelihoods of coastal citizens. This study focuses on answering the following questions: which scenarios are most likely in coastal areas? What kinds of dynamics would drive such a transition in coastal areas? How could the potential circular tourism economy be most effectively adopted?

## 2. Literature Review

In the European economy, small enterprises play a vital role because they can lead to the creation of family businesses that provide a diversified source of income and contribute to national economic value [29]. However, small enterprises often do not have enough cash flow to cover capital costs and high costs for other business models, which means they must rely on the sharing economy as an alternative—sharing instead of buying can help small businesses to survive [30]. The sharing economy concept has been positioned around three fundamental cores: (1) access economics: sharing property benefits to optimize consumption; (2) the platform economy: optimizing technology as a tool for consumption exchange on digital platforms; and (3) a community-based economy: based on the concept of win–win cooperation [31]. Other concepts, such as "sustainability", are frequently used as synonyms for a sharing economy that promotes sustainable consumption rather than purely market-based exchanges; sustainability leads to a reduction in net consumption and resource use [25,32]. There is lower productivity in the absence of resources sharing [32]. Examples of the sharing economy, e.g., bike-sharing systems and homestay accommodation, have emerged in urban and rural areas. As defined by Kozak [33], the substance of the concept of integrated development is not only the revival and regularization of these areas

but also the improvement of the quality of life as a whole. Specifically, in Finland and China, demand for short-term rentals is emerging on multiple digital platforms. Activities aimed at producing cultural value were encouraged in Helsinki, Finland, through small-scale entrepreneurship, facilitating Airbnb's growing popularity [34]. In Shanghai, China, tourists are beginning to prefer Roomorama and the Airbnb platform to traditional hotels: the total sales and user base of these accommodation-sharing platforms has increased significantly, reaching 16.5 billion and 147 million, respectively, in 2018 [35,36].

One strategy some coastal families are beginning to explore is the addition of value from the ecotourism industry. This can take the form of farm tours, fishing tours, or fruit-picking tours on organic farms, or offering wilderness experiences and leisure activities, which are an indispensable core value in tourism. In developing countries such as Vietnam, promoting the economic benefits of tourism while preserving ecological sustainability and sociocultural heritage is a constant challenge [37,38]. Several studies have proposed ecotourism to secure economic advantages from the tourism sector while conserving environmental resources and protecting sociocultural heritage through ecotourist activities aimed at creating values of common benefit [39–41]. In particular, the cooperation and partnership of all stakeholders in this process create a circular economy that is related to well-being and local ecosystem functions. In some cases, ecotourism is viewed as an effective tool for preserving cultural value and conserving natural resources [39,42,43]. However, there are also situations where ecotourism has exacerbated the destruction of local natural resources; this can be attributed to a number of factors, including a lack of cooperation and unfair competition between stakeholders in terms of strategic planning and practical operations [44,45]. Local residents' awareness is also crucial for sustainable tourism development in terms of modifying consumer behavior [46,47].

For this purpose, different from traditional accommodation models, the Airbnb platform can be integrated with ecotourism to create accommodation on a working farm. This concept benefits from the rational exploitation of local resources, flexible working times, lower operation costs for coastal residents, lower prices for tourists, and the fact that Airbnb hosts can interact with tourists through daily activities in their home, offering guests opportunities to experience "one day as a farmer" or fishing experiences, as well as learning how a farm functions. Volgger et al. [48] claimed that Airbnb users have a keen interest in exploring known tourist highlights of the destination. Thus, Airbnb users are looking for a profoundly "alternative travel destination", driven by the motivation to seek novel experiences compared to other tourist groups and value the relevancy and authenticity of the Airbnb community [49]. However, another perception revealed some potential issues with the sharing economy, such as a fear of strangers and social problems caused by interactions between local residents and tourists [50]. Cadoret [51] also studied how to enhance the important role of stakeholders in establishing a protection policy and transforming local living spaces with a new business model. However, these issues as inevitable pressures were driven by digital technologies, which create a sustainable economy for human well-being [52].

As per the above research, we propose that the circular economy has positive economic effects in terms of coastal development.

**Hypothesis 1 (H1).** *The integration of the sharing economy and ecotourism has positive economic effects (PEE) on coastal development.*

**Hypothesis 2 (H2).** *The integration of the sharing economy and ecotourism creates positive economic pressure (PEP), benefiting coastal development.*

Many studies have agreed that the role of sustainable consumption in the sharing economy is to be efficiently directed towards the use of local resources and respecting others' well-being in terms of production and consumption processes [53–55]. Traditional sustainable consumption is defined by Wang et al. [56] as personally motivated consumption, as well as what is beneficial for individuals' consumption. However, in the new

sharing economy, studies have examined sustainable consumption behaviors with the effective support of technology [54,57,58]. For example, sharing economy models of house sharing and sharing bicycles on the forums of Airbnb and Didi have been analyzed in terms of opportunities and challenges, pointing to a circular economy that reflects the relationship between supply and demand for the mutual benefit of stakeholders [59–61]. This concept of sustainable consumption has also been reshaped toward the goal of sustainable development (SDG), which involves minimizing the use of natural resources and the emission of waste and pollutants at the macro and micro levels based on circular economic activities [55].

Therefore, as a technological intermediary, the decision to use Airbnb was based on elements such as sustainability, enjoyment, and economic benefits for those who want to rent out space and those who are looking for a space to rent [62]. Technology intermediaries receive a commission from consumers, which is included in the price of the service. It is also a peer-to-peer service, meaning that customers and service providers are interchangeable at different times [63]. More specifically, service providers can run an Airbnb where they live but can also become customers when they wish to find a service provider in another location. Consequently, the benefits are shared—customers find a cheaper price and more affordable accommodation, different from traditional accommodation, and renters can earn extra income through their available space [64].

Compared to the hospitality industry's traditional accommodation services, Airbnb's service is recognized for its use of private spaces and its innovative design that makes tourists feel more relaxed and excited. In a survey study by Mody et al. [65], it was found that the hospitality in hotels is no better than that at an Airbnb, meaning that guests prefer the Airbnb platform for enjoyable experiences, which is considered to be one of the factors leading to competition with the local hotel system. Guttentag [23] found that the Airbnb model had a negative impact on hotel revenue because the price on Airbnb was cheaper than at local hotels, which resulted in most hotels in Texas, USA having to lower their rates. Consequently, government agencies and hotels also treat Airbnb as a threat to hospitality businesses and traditional tourism [66]. In an Airbnb, sharing a living space with strangers can disrupt the daily life of locals, especially for foreigners with different customs and education levels [67,68]. Simultaneously, the growth of the residential service industry has meant an increase in the prices of essential goods, although not all residents are involved in the sharing economy [69].

For these reasons, the following hypotheses were drawn up to assess the economic benefits of Airbnb and its effects on living conditions in coastal zones:

**Hypothesis 3 (H3).** *Airbnb offers economic benefits that positively improve residents' living conditions in coastal zones (IRLC).*

**Hypothesis 4 (H4).** *Airbnb brings economic benefits that put pressure on the living conditions of local residents in coastal zones (PRLC).*

Community-based ecotourism is a form of tourism that involves a sense of responsibility for preserving the local culture and natural settings. Masud et al. [70] claim that this model leads to sustainable economic growth by improving local living conditions. This helps to foster local participation in the preservation of ecosystem resources and a friendly community environment. Nevertheless, local residents can only benefit from the proposed sustainable development if local policymakers and shareholders work towards a common goal of human well-being [47]. These factors contribute significantly to the success of cooperative policies and minimizing the negative effects on local development [71].

In other words, locals and tourists share the benefits of resources and contribute to them through knowledge, money, and services [72]. Wu and Shen [64] illustrated that Airbnb platforms can serve as a reference for the worldwide sustainable development of the sharing economy by instilling cultural values in different communities that foster regional development. For example, in Guilin, China, famous for its ancient villages, the local Government teamed up with Airbnb and took advantage of rural villages to create a

unique culture to attract tourists. This project contributed to supporting the income of rural people, helping them to escape poverty, with the sharing economy model as a key source of revenue. Another typical example is the ancient town of Hoi An in Vietnam. Since the advent of Airbnb, this neighborhood has become famous for its lantern-lit streets at night, which facilitates the process of preserving long-standing traditional cultural values. The living conditions of the people of this old town began to be improved, and cultural values were preserved due to the efforts of the local Government. Nevertheless, because the economic sharing business model is based on technology platforms, the growing popularity of social networks and technology is the most powerful feature that drives the sharing economy due to coastal residents' lack of professionalism and technological sophistication. Many of them cannot speak a foreign language, which creates a challenge in the training process. For example, many farmers are unable to adhere strictly to market standards due to a lack of training programs and the requisite technological capacity in most developing countries [73].

We further contend that the integration process of Airbnb and coastal farmers is related to the question of trust between the exchanging parties, as has also been indicated in previous studies [64,67,74]. The hosts on Airbnb must be open to meeting strangers from different countries; thus, trust plays a vital role in the Airbnb platform because of sharing and the fact that the financial transaction is mediated by a third party with technological assurances. Tussyadiah and Pesonen [75] posited that collaborative consumption means trusting strangers, and a lack of knowledge and ability to use the service may be perceived as constraints on adopting peer-to-peer accommodation systems.

In the context of Airbnb, several researchers have analyzed the crucial role of trust in the Airbnb community network, emphasizing the key function of the sharing economy mode through identity verification [76]. In fact, these identity-related issues can be ensured by the reputation mechanisms of monetary transaction parties and local government support [77]. Others suggest that there are negative impacts to neighborhood living, including noise pollution and cultural differences in urban areas [78]. Thus, it is hypothesized that local residents expected Airbnb to support local sustainable development. A questionnaire with a three-item scale will be used, with questions such as "I prefer to take full advantage of my home space to participate in the Airbnb farm stay service" to measure local residents' attitude towards Airbnb in coastal zones.

**Hypothesis 5 (H5).** *Residents believe that Airbnb can positively impact sustainable development in coastal zones (P-SSDC).*

**Hypothesis 6 (H6).** *Residents believe that Airbnb has negative effects on sustainable development in coastal zones (N-SSDC).*

In this study, we combined economic influences/impacts into one variable that was assessed through four factors: positive economic effects (PEE), positive economic pressure (PEP), expected improvement in residents' living conditions due to Airbnb (IRLC), and pressure on residents' living conditions due to Airbnb (PRLC) (see Figure 2). The hypotheses IRLC and PRLC were studied as keys to resolving the correlation between Airbnb and ecotourism influences in coastal zones as moderator variables.

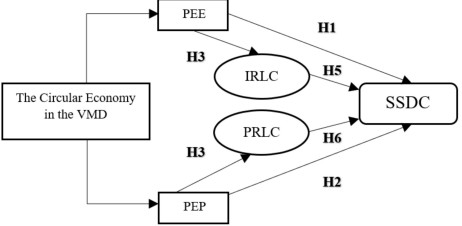

**Figure 2.** The proposed conceptual model. PEE, positive economic effects; PEP, positive economic pressure; IRLC, expected improvement in residents' living conditions from Airbnb; PRLC, pressure

on residents' living conditions from Airbnb; SSDC, support for sustainable development in coastal zones (P-SSDC, positively impact sustainable development in coastal zones; N-SSDC, positively impact sustainable development in coastal zones); H1–H6, Hypotheses 1–6 of the knowledge and ability to use the service may be perceived as a constraint in adopting peer-to-peer accommodation systems.

## 3. Study Area and Methods

### 3.1. Study Area

The Mekong River stretches across China, Myanmar, Thailand, Cambodia, and Vietnam, creating fertile regions that are accessible to residents. The Mekong Delta is the largest rice and tropical fruit production area in Vietnam. Beautiful natural destinations and historical destinations are some of the potential resources of the Vietnam Mekong Delta, but the linking of local ecological and tourist attractions has not been emphasized. Ecosystems in the Vietnam Mekong Delta are extremely diverse and need to be conserved regularly and continuously [79]. The Mekong Delta accounts for only 12% of the area of Vietnam. From a national economic perspective, this area plays a key role in providing more than 50% of the country's rice, 26.1 million tons of rice, 48% of the cereal production, 75% of the aquaculture production, 38% of the marine fisheries, and 40% of the fish caught compared to the total Vietnamese production in 2018 [80].

In this paper, the authors argue that ecosystem diversity and historical destinations are among the most valuable resources of the Vietnam Mekong Delta [81,82], but the integration of the sharing economy has not been emphasized by policymakers. We suggest that there is ample value to be gained from directing institutional resources to sustainable development depending on the coastal natural resources. A comprehensive economic perspective on coastal resource values is needed, in which the benefits derived from ecosystem services must be recognized and play an important part in coastal management policies. Therefore, this research takes the Vietnam Mekong Delta, shown in Figure 3, as the study area in order to explain coastal residents' attitudes towards the integration of Airbnb and ecotourism.

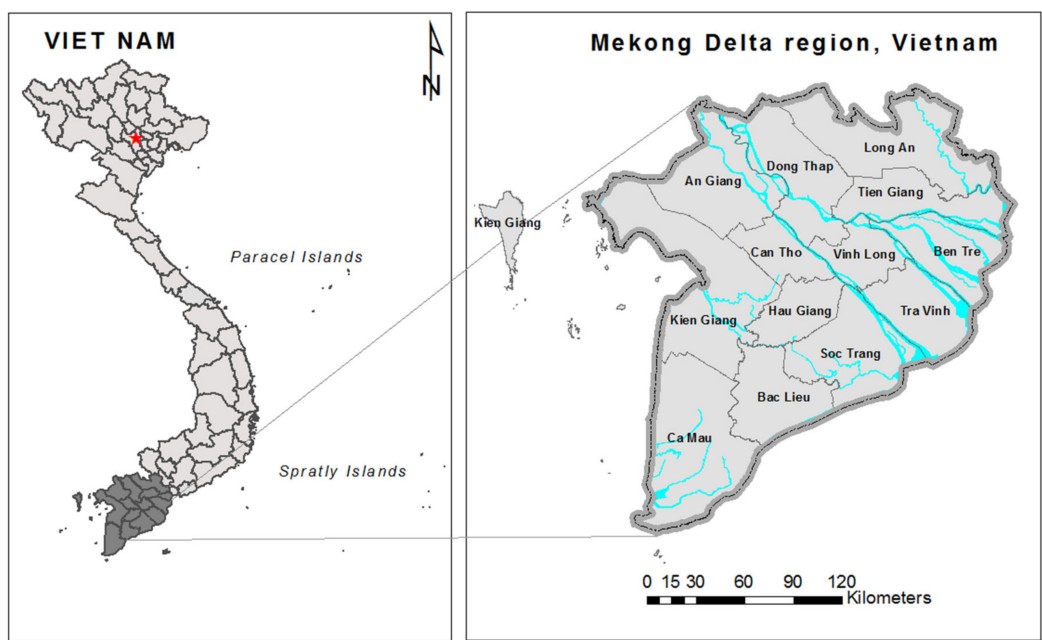

**Figure 3.** The Vietnam Mekong Delta (VMD) is a peninsula with three edges—northeast, southeast, and west—bordering the sea (with a coastline of 700 km). Sources: [83].

### 3.2. Method

In this study, the hypotheses were tested using Amos 20. We applied a structural equation model to analyze the relationship between dependent and independent variables.

Confirmatory factor analysis (CFA) and structural equation modeling (SEM) were briefly applied to test the overall relationship between factors in the proposed model [84]. For the purpose of this study, coastal residents from 12 provinces in the Vietnam Mekong Delta were invited to fill out a questionnaire. The content of the questionnaire was easy to understand, simple, and concise. The survey was conducted from 10 August to 15 December 2020, with the support of the Mekong Delta Business Association. The residents participated in the survey enthusiastically and responsibly; respondents had no conflict of interest with researchers, and written informed consent was obtained from them.

The questionnaire had three parts and a total of 26 questions (Appendix A). Table A1 gives the basic screening data. The purpose of the first section was to classify the participants by income, gender, and place of residence. If the questionnaire survey is aimed at the right audience, the data will be more accurate. Table A2 examined a total of 20 phrases, classified into two subgroups, to measure the structural relationships between the variables IRLC, PRLC, and SSDC.

## 4. Results

### 4.1. Social Demographics and Descriptive Results of the Respondents

We used a five-point Likert scale, and 26 questions were asked through a cross-sectional survey. According to the survey, 46.9% were females, and 53.1% were males. The respondents can be considered evidence of an aging population: 35.6% were over 46 years old, and 29.4% were between 26 and 45 years old. As many as 49.2% of the residents of the Vietnam Mekong Delta worked in aquaculture, and 4.6% had started an Airbnb business. Most respondents (37.3%) had an income of 551–699 USD/month, mainly concentrated in private businesses. One of the most important sectors contributing to the development of the region is agri-aquaculture; farmers often have an income of 350–500 USD/month in the harvest season, accounting for 22.8% of respondents. There were also a number of residents who had no income or low income (150 USD/month), comprising 4% and 7.6%, respectively. They mostly worked as hired laborers for large farms, as shown in Table 1.

**Table 1.** Respondents' demographics variables.

| Demographics | Item | Subjects (*n* = 303) | |
|---|---|---|---|
| | | **Frequency** | **Percentage (%)** |
| Gender | Male | 161 | 53.1 |
| | Female | 142 | 46.9 |
| Age | Less than 25 | 42 | 13.9 |
| | 26–35 | 64 | 21.1 |
| | 36–45 | 89 | 29.4 |
| | More than 46 | 108 | 35.6 |
| Income (USD) | No Income | 12 | 4 |
| | <150 | 23 | 7.6 |
| | 151–349 | 26 | 8.6 |
| | 350–500 | 69 | 22.8 |
| | 551–699 | 45 | 14.9 |
| | 700–899 | 113 | 37.3 |
| | >900 | 15 | 5 |
| Ethnic Group | Kinh | 199 | 65.7 |
| | Hoa | 28 | 9.2 |
| | Khmer | 15 | 5 |
| | Cham | 40 | 13.2 |
| | Others | 21 | 6.9 |
| Occupation | Agri–aquaculture | 149 | 49.2 |
| | Tourism | 27 | 8.9 |
| | Family Business | 78 | 25.7 |
| | Airbnb | 14 | 4.6 |
| | Government Office | 11 | 3.6 |
| | Others | 24 | 7.9 |

The descriptive results of the questionnaire survey are statistically depicted to understand coastal residents' attitudes towards the circular economy of the sharing economy. Relating to positive constructs, the survey report illustrated that coastal residents in the Vietnam Mekong Delta had high expectations for the potential development benefits of Airbnb. The mean score of all items (on a five-point scale) of the two positive constructs, PEE and IRLC, were 4.33 and 4.27, respectively, as listed in Table 2. More than 80% of respondents agreed or strongly agreed with the items relating to the item-scale SSDC (support for sustainable development in coastal zones). For example, 34.7% and 46.2% of respondents agreed and strongly agreed, respectively, with the statement, "I prefer to take full advantage of a coastal ecosystem to improve coastal sustainable development." There was a similar result for the item "I am willing to be involved in the sharing economy to foster ecotourist-based coastal sustainable development." High expectations for the VMD communities were also recorded: 37.6% agreed, and 45.9% strongly agreed. Most of these responses were from those who work in the tourism industry, a family business, or the Airbnb sector, in the belief that this integration could lead to positive benefits through the sharing economy and ecotourism.

**Table 2.** Residents' attitudes about integrating the sharing economy through the Airbnb model in the VMD, measured through AMOS 20.

| Construct | Strongly Disagree (%) | Disagree (%) | Neutral (%) | Agree (%) | Strongly Agree (%) | Mean | Std. Deviation | Factor Loading | Composite Reliability | Average Extracted Variance |
|---|---|---|---|---|---|---|---|---|---|---|
| Expected Positive Economic Effects (PEE) | | | | | | | | | 0.804 | 0.58 |
| PEE1 | 0.30% | 4.00% | 12.90% | 32.30% | 50.50% | 4.29 | 0.861 | 0.757 | | |
| PEE2 | 0.30% | 3.30% | 13.50% | 27.10% | 55.80% | 4.35 | 0.859 | 0.749 | | |
| PEE3 | 0.30% | 3.60% | 10.90% | 31.00% | 54.10% | 4.35 | 0.84 | 0.721 | | |
| Positive Improvement on Residents' Living Condition from Airbnb (IRLC) | | | | | | 4.27 | | | 0.801 | 0.574 |
| IRLC1 | 0.00% | 3.30% | 13.20% | 35.30% | 48.20% | 4.28 | 0.817 | 0.699 | | |
| IRLC2 | 0.00% | 3.30% | 14.90% | 34.00% | 47.90% | 4.26 | 0.832 | 0.778 | | |
| IRLC3 | 0.00% | 3.60% | 13.90% | 35.60% | 46.90% | 4.26 | 0.83 | 0.76 | | |
| IRLC4 | 3.60% | 4.60% | 14.20% | 35.00% | 42.60% | 4.08 | 1.037 | x | | |
| Pressures on Residents' Living Conditions from Airbnb (PRLC) | | | | | | | | | 0.861 | 0.508 |
| PRLC1 | 18.50% | 35.00% | 30.00% | 13.50% | 3.00% | 2.48 | 1.035 | 0.649 | | |
| PRLC2 | 17.20% | 34.70% | 29.70% | 14.50% | 3.00% | 2.52 | 1.029 | 0.66 | | |
| PRLC3 | 16.80% | 41.30% | 24.40% | 26.40% | 1.00% | 2.46 | 1.028 | 0.701 | | |
| PRLC4 | 21.10% | 30.00% | 31.00% | 15.50% | 2.30% | 2.48 | 1.06 | 0.766 | | |
| PRLC5 | 21.10% | 33.70% | 28.10% | 13.90% | 3.30% | 2.45 | 1.072 | 0.69 | | |
| PRLC6 | 19.80% | 32.00% | 31.40% | 13.50% | 3.30% | 2.49 | 1.057 | 0.769 | | |
| Positive Economic Pressures (PEP) | | | | | | | | | 0.804 | 0.555 |
| PEE1 | 4.30% | 40.30% | 32.00% | 18.20% | 5.30% | 2.8 | 0.964 | 0.802 | | |
| PEE2 | 10.20% | 34.00% | 34.00% | 16.50% | 5.30% | 2.73 | 1.027 | 0.761 | | |
| PEE3 | 5.90% | 33.30% | 38.60% | 17.80% | 4.30% | 2.81 | 0.943 | 0.573 | | |
| Support for Sustainable Development in Coastal Zones (SSDC) | | | | | | | | | 0.8 | 0.501 |
| SSDC1 | 0.00% | 4.60% | 14.50% | 34.70% | 46.20% | 4.22 | 0.863 | 0.658 | | |
| SSDC2 | 0.00% | 4.30% | 17.20% | 30.70% | 47.90% | 4.22 | 0.881 | 0.739 | | |
| SSDC3 | 0.70% | 4.00% | 11.90% | 35.30% | 48.20% | 4.26 | 0.867 | 0.725 | | |
| SSDC4 | 0.30% | 5.60% | 10.60% | 37.60% | 45.90% | 4.23 | 0.876 | 0.689 | | |

Note: x represents nonsignificant loading.

In terms of the VMD communities' attitude about the two pressure constructs, PEP and PRLC, the mean score was 2.77 and 2.48, respectively, illustrating that most local residents disagreed that this circular economy of the sharing economy would have a positive influence on their living conditions. More specifically, 40.3% and 4.3% of respondents

disagreed and strongly disagreed, respectively, with the item "the sharing economy will increase infrastructure costs" in the PEE construct; however, a neutral response was also given by 32% of respondents, indicating that coastal residents have not decided whether the pressure caused by the sharing economy will be positive or negative. For all scale items of the PRLC construct, most respondents expressed disagreement or a neutral response, with 35% and 30% of the respondents who work mainly in the Agri-aquaculture sector disagreeing or remaining neutral, respectively, on the scale item "Airbnb will allow strangers to cause disturbances in my daily life." The findings indicate that residents have different ideas about the influence the sharing economy will have on the circular economy in their community.

A further analysis of reliability will be conducted to determine the association between scale items from different constructs.

### 4.2. Reliability Analysis

The test results for reliability based on the Cronbach alpha coefficients illustrate that all the components of the inspection scale were acceptable (see Table 2). The composite reliability of the questionnaire survey was tested with Cronbach's $\alpha$, with values between 0.800 and 0.861, indicating that all the proposed constructs have high reliability. In theory, the higher the Cronbach's $\alpha$, the better (a high score on the confidence scale). This does not happen in many cases. If the Cronbach's $\alpha$ coefficient is too large, many variables on the scale will exhibit no difference, considering that the variable total correlation coefficient (adjusted) of the observed variables meets the requirement of being >0.30 [85]. Based on this principle, the IRLC4 variable (Airbnb will cause social problems, including alcoholism, that negatively affect the VMD community) needs to be removed because the corrected item has a total correlation of $\leq 0.3$. Therefore, after the IRLC variable (Pressures on Residents' Living Conditions from Airbnb) was retested, the scale was deemed suitable for use in exploratory factor analysis.

### 4.3. Exploratory Factor Analysis (EFA)

All observed variables met the requirements for the exploratory factor analysis. The factor used is principal axis factoring (PAF) with no perpendicular rotation in Promax. The EFA results had a KMO coefficient of 0.881, which is a good figure; the Bartlett test value was significant at <0.05. Five groups of factors were extracted with a total error of 54.43%. Therefore, the scale of hypotheses after the preliminary assessment included five components—(1) PEE, (2) IRLC, (3) PRLC, (4) PEP, and (5) SSDC—with 19 observed variables analyzed in the factor analysis (CFA).

### 4.4. Confirmatory Factor Analysis (CFA)

To evaluate the convergence validity, a confirmatory factor was used to identify potential factors, calculated with variation and covariation against sets of measurement variables [86], and a structural equation modeling was used to test the associations between the proposed constructs (see Figure 4). The coefficient of correlation between the components with the attached standard deviation (see Table 2) reflects that the weights of the observed variables meet the permitted standard ($\geq 0.5$), and statistically significant *p*-values were equal to 0.000, indicating that the variables were all suitable for use as distinguishing factors. Therefore, we can conclude that the observed variables used as measures in the survey achieved the convergence value. A measurement model fit was performed using AMOS 20 to test the model fitness, in which the $\chi 2/df$ value was less than 2 (1.185); the root means square error of approximation (RMSEA) was less than 0.06 (0.025); the goodness of fit (GFI) and comparative fit index (CFI) were more than 0.9 (0.946 and 0.988, respectively), and the non-normed fit index (NNFI) or TLI was 0.986, indicating that all reached the threshold value.

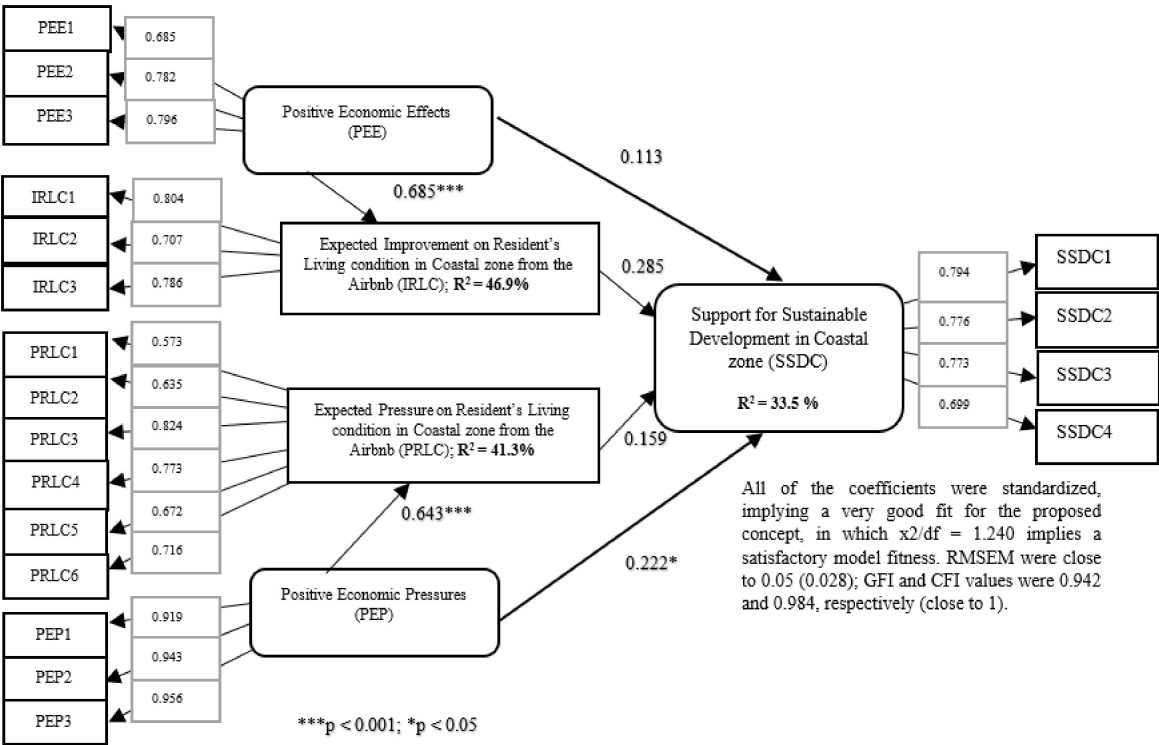

**Figure 4.** Proposed model of coastal residents' expectations for circular economy between the sharing economy and ecotourism in the Vietnam Mekong Delta.

### 4.5. Relationships between Proposed Constructs

Mediation analyses were also conducted to examine the relationships among the hypotheses (see Figure 5). The figure shows that PEE had a positive correlation with IRLC but not with P-SSDC, reflecting that the positive economic effects expected by local residents do not directly lead to support for sustainable coastal development in Vietnam Mekong Delta. However, there was a positive correlation between IRLC and P-SSDC, meaning that IRLC serves as an important predictor in support of sustainable development. Therefore, Hypotheses H3 and H5 were supported.

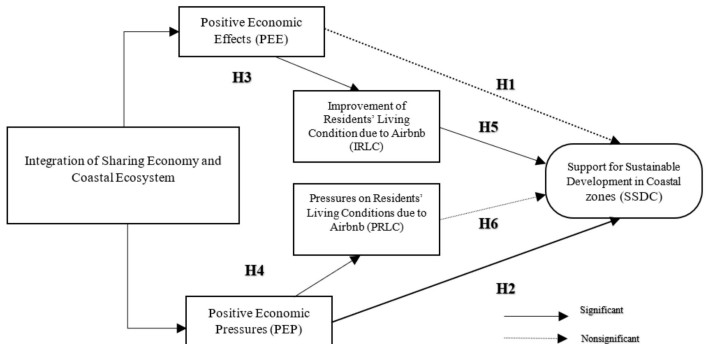

**Figure 5.** Relationships between coastal residents' expectations and coastal circular-economy-based sustainable development in the Vietnam Mekong Delta.

In contrast, the figure suggests that PEP was positively associated with PRLC and N-SSDC, indicating that coastal people believe the circular economy in their communities will result in economic pressures such as they have never experienced, leading to negative attitudes and supporting Hypotheses H2 and H4. It is noteworthy that PRLC did not have

a significant effect on the N-SSDC, leading to the PRLC not being a meaningful mediator of local residents' support for sustainable development in the Vietnam Mekong Delta.

Therefore, based on the analysis mentioned, it can be concluded that IRLC plays an important mediating factor in the coastal circular economy (as was postulated in Hypotheses H3 and H5). Meanwhile, PRLC does not directly affect sustainable coastal development but, instead, PEP has a direct positive relationship with PRLC and N-SSDC (as postulated in Hypotheses H4 and H2). Hypotheses H1 and H6 were rejected.

## 5. Discussion

With regard to the method applied to the Vietnam Mekong Delta, understanding the attitude of coastal locals towards Airbnb (in terms of barriers and potential opportunities) is an important experimental research topic in coastal zones. Local authorities and investors should seek to gain a community-based view as a reference to adjust their plans for coastal development.

The findings of this study show that local residents of the Vietnam Mekong Delta have high expectations for the innovation of the Airbnb model because they expect that the integration of the sharing economy and ecotourism can support coastal residents' livelihoods. This study also provides an in-depth understanding of the role of local occupation in supporting coastal development. Although the local residents have not really participated in the Airbnb farm stay model (only 4.6% of respondents run an Airbnb business), they believe that the ecotourism-based Airbnb model could improve their living conditions. The difference between a traditional Airbnb and the new model is the involvement of stakeholders and local authorities in order to connect individual resources to create mutual benefit for the community's well-being. Therefore, hosts using Airbnb can easily identify the influences of the circular economy of the sharing economy and local ecotourism. A number of respondents working in tourism or a family business strongly agreed about the improvement Airbnb can create in residents' living conditions, illustrating their high expectations for coastal integration. Meanwhile, economic pressure is the major concern for respondents who work for the Government. Those in Agri-aquaculture-based occupations had a neutral attitude about the sharing economy's integrated model. These findings may help policymakers develop appropriate strategies for each local career type to increase efficiency as well as improve communication with residents.

To facilitate sustainable tourism development, the Government of Vietnam, since 2010, and in collaboration with NGOs, has started tourism projects to enhance benefits for local communities [78]. A large project with the support of JICA (the Japan International Corporation Agency) was established in the Mekong Delta to provide water resource-management technology for the community and improve people's lives as well [78]. Additionally, as a tourism development strategy associated with the preservation of cultural values of the Vietnamese Government, the orientation framework "Development of the Mekong Delta in the period of 2021–2030 with a vision to 2050" identified the Mekong Delta as having many potential development advantages. For example, typical island tourism on Phu Quoc Island (Kien Giang), spiritual tourism, e.g., sightseeing at An Giang, or cultural tourism, e.g., festivals in Ca Mau, Bac Lieu, and Soc Trang provinces [15]. Based on the tourism-based development orientation of that project, the Mekong Delta is assessed as having potential for development based on its rich resources. Here, cooperation between tourists and locals is emphasized. Traditional lodging services have been highlighted in relation to resort tourism and high-class sightseeing. Additionally, the Mekong Delta is recognized as having the ability to attract tourists because of being authentic and rustic, attracting tourists who like the experience and form of Airbnb ecotourism [86].

However, in the Vietnam Mekong Delta, ecotourism, a specific tourist model aimed at raising environmental awareness, is facing many infrastructure challenges due to insufficient investment in transportation and human resources, resulting in a lack of quality and uniqueness in tourist products. Tourists prefer short-term "1-day" tours, which results in low revenue for all stakeholders. Despite the fact that tourist resources are diverse, with

many typical experiences such as picking fruit, boating in canals, listening to folk music, and participating in traditional village crafts, the Vietnam Mekong still faces environmental pollution problems due to tourism.

## 6. Conclusions

This study provides a potential theoretical framework for coastal sustainable development researchers. First, because coastal zones possess complex and unique characteristics, the national government is less inclined to support strategic economic development projects. This study focused on the extension of Airbnb into coastal zones. This contribution represents the first attempt to analyze the integration of the sharing economy and local ecotourism in coastal zones for an in-depth understanding of the key effects on sustainable coastal development in a circular tourism economy. Moreover, with the aim of achieving Sustainable Development Goals (SDGs), a community-based approach plays a significant role in understanding coastal residents' attitudes toward poverty reduction. As such, implementing sustainable development can be fostered by developing residents' trust in the government, leading to a good relationship with coastal residents in any further projects.

Second, based on coastal resources, the role of stakeholders is prioritized in sustainable development because of their important role in the cooperation process, along with supporting policies from the government. Private economic sectors can be promoted not only for economic purposes but also for environmental preservation and social values. Our results illustrate the integration of coastal development with Airbnb and ecotourism in the Vietnam Mekong Delta. Therefore, future research should be conducted on how to develop and improve coastal development, not only from a circular economic perspective, but also in terms of preserving local cultural values, as well as the environment, thereby fostering Sustainable Development Goals (SDGs). As a result of resource constraints and limited time, the sample size may represent only some of the extant opinions on this matter. Research should be conducted with a larger sample size and using more modern techniques and methods in the future. This study may also suggest other research directions for sustainable development and agricultural development in coastal zones.

Therefore, based on our results, we propose a transition of traditional tourism towards a circular economy by applying the theory of the sharing economy and ecotourism in three dimensions:

First of all, acknowledging that the 2030 Agenda for Sustainable Development Goals is a core value of national development, as featured by the integration of Airbnb and ecotourism to create a tourism-based circular economy in an effort to improve residents' living conditions in coastal zones. Our findings suggest that different business sectors involved in ecotourism activities may develop from the integration of coastal resources by utilizing unused living space, as well as existing ecotourism activities, such as traditional village crafts, fruit picking, fishing, and farming.

Secondly, understanding the role of coastal resources is important in achieving the goal of sustainable coastal development, which can gradually hand over authority to coastal residents for local value preservation. Within such a framework, coastal resource-based practices can be linked into a new form of sustainable coastal development that results from a local system-wide co-business wherein responsibilities and rights are the guiding principle for all stakeholders. Finally, the proposed approach emphasizes the role of coastal policymakers in the integration and complex compound practices through technology and language-ability training programs. Through empirical research in the Mekong Delta, we found that there is little consistency among respondents regarding local activities, including opportunities and challenges. This may point to a potential role for policymakers in terms of opening suitable paths for local development; this is an area not currently highlighted in coastal research.

Overall, our findings indicate that coastal residents' living conditions, as a moderating factor to explore the impact on coastal sustainable development, will face challenges as a result of the COVID-19 pandemic, as well as changes in existing coastal resources.

This positive change needs to be promoted by a suitable path, followed by Sustainable Development Goals (SDGs). The new form of the circular tourism economy, if it proves capable of adapting to coastal residents' expectations, may be a possible solution for sustainable coastal development.

**Author Contributions:** Conceptualization, J.H.; methodology, J.H. and T.H.T.M.; software, T.H.T.M.; writing—original draft preparation, T.H.T.M.; writing—review and editing, T.H.T.M.; visualization J.H.; supervision, J.H. All authors have read and agreed to the published version of the manuscript.

**Funding:** This research was funded by the National Natural Science Foundation of China, grant number 71871144; Science and Technology Development Project of the University of Shanghai for Science and Technology, grant number 2020KJFZ046.

**Institutional Review Board Statement:** Not applicable.

**Informed Consent Statement:** Informed consent was obtained from all subjects involved in the study.

**Data Availability Statement:** Data available on request due to restrictions, i.e., privacy.

**Conflicts of Interest:** The authors declare no conflict of interest.

## Appendix A. Questionnaire on the Residents' Attitudes toward Circular Economy in the Vietnam Mekong Delta

**Table A1.** Personal information.

| | |
|---|---|
| Gender | ☐ M ☐ F |
| Age | ☐ ≤25 ☐ 26–35 ☐ 36–45 ☐ ≥45 |
| Hometown | ☐ An Giang, Kien Giang, Bac Lieu, Ca Mau<br>☐ Tra Vinh, Hau Giang, Soc Trang, Dong Thap<br>☐ Long An, Tien Giang, Ben Tre, Vinh Long |
| Ethnic groups | ☐ Kinh<br>☐ Hoa ☐ Khmer ☐ Cham<br>☐ Other |
| Monthly Salary (USD) | ☐ No income ☐ <150 USD ☐ 151–349 USD ☐ 350–500 USD<br>☐ 551–699 USD ☐ 700–899 USD ☐ >900 USD |
| Occupation | Agri-aqua production<br>Tourism<br>Family business<br>Airbnb<br>Government office<br>Others |

**Table A2.** Residents' attitudes towards the circular economy of the sharing economy on the Airbnb farm stay model in the Vietnam Mekong Delta.

| Items | Description | Strongly Disagree (SD) | Mildly Disagree (MD) | Neutral (N) | Mildly Agree (MA) | Strongly Agree (SA) |
|---|---|---|---|---|---|---|
| PEE1 | 1. The sharing economy will promote coastal development through ecotourism in the VMD. | | | | | |
| PEE2 | 2. The sharing economy will create employment opportunities such as tour guide, food services, housekeeper, etc. | | | | | |
| PEE3 | 3. The sharing economy will generate extra income from shared accommodations for the VMD residents. | | | | | |

**Table A2.** *Cont.*

| Items | Description | Strongly Disagree (SD) | Mildly Disagree (MD) | Neutral (N) | Mildly Agree (MA) | Strongly Agree (SA) |
|---|---|---|---|---|---|---|
| IRLC1 | 4. Airbnb will enhance local cultural exchange if local cultural values are incorporated such as home decor, souvenirs, and activities with local residents. | | | | | |
| IRLC2 | 5. Airbnb will improve residents' knowledge when participating in operating the Airbnb model in the VMD. | | | | | |
| IRLC3 | 6. Airbnb will assist in conserving local ecosystems through linking to ecotourism in the VMD. | | | | | |
| IRLC4 | 7. Airbnb will reduce environmental impacts (e.g., energy efficiency, food waste) on the VMD community. | | | | | |
| PRLC1 | 8. Airbnb will allow strangers to cause disturbances in my daily life. | | | | | |
| PRLC2 | 9. Airbnb will cause conflicts between coastal accommodation services and Airbnb hosts, leading to a highly competitive market. | | | | | |
| PRLC3 | 10. Airbnb will increase noise pollution, affecting the local environment in the VMD community. | | | | | |
| PRLC4 | 11. Airbnb will cause social problems, including alcoholism, that negatively affect the VMD community. | | | | | |
| PRLC5 | 12. Airbnb will increase crime/robberies in the VMD. | | | | | |
| PRLC6 | 13. Airbnb will cause difficulties in communication due to diverse cultures and different languages. | | | | | |
| PEE1 | 14. The sharing economy will increase infrastructure costs. | | | | | |
| PEE2 | 15. The sharing economy will increase the cost of preserving ecosystem value. | | | | | |
| PEE3 | 16. The sharing economy will lead to uniformity of the ecological structure. | | | | | |

Note: VMD: Vietnam Mekong Delta.

**Table A3.** Willingness to use the Airbnb farm stay service in your home with the integration of coastal resources for local development in the Vietnam Mekong Delta.

| Items | Description | Strongly Disagree (SD) | Mildly Disagree (MD) | Neutral (N) | Mildly Agree (MA) | Strongly Agree (SA) |
|---|---|---|---|---|---|---|
| SSDC1 | 17. I prefer to take full advantage of the coastal ecosystem to improve coastal sustainable development. | | | | | |
| SSDC2 | 18. I am happy to introduce local culture to tourists. | | | | | |
| SSDC3 | 19. I expect the ecological value of the coastal zones to be preserved. | | | | | |
| SSDC4 | 20. I am willing to be involved in the interconnection of the sharing economy to foster ecotourist-based coastal sustainable development. | | | | | |

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
