# Peer review of "The Circular Economy: A Study on the Use of Airbnb for Sustainable Coastal Development in the Vietnam Mekong Delta"

_sustainability, doi:10.3390/su13137493_

Round 1

Reviewer 1 Report

Please use more references.

https://www.academia.edu/Documents/in/Airbnb

Shorter title is recommended (Coastal word is duplicated).

Coastal Interconnection: A Study on the Use of Airbnb for Coastal Sustainable Development in the Vietnam Mekong Delta

Author Response

Dear Reviewer,

Thank you for your feedback; many references have been cited to improve contextualization with regard to the previous and current theoretical background and empirical research. Literature reviewer, hypotheses, and methods of research are also added to clarify the research topic.

After revising the reviewer's comments, the topic was reshaped and shortened to reflect a new type of circular economy. I am grateful for your review.

Reviewer 2 Report

My only major comment was the use of the map for the Mekong Delta and its map inset. The map is missing many major map elements (such as source notes, data source, and author) and the inset does not cartographically match the larger map. The conceptual diagrams are nicely detailed and polished, which makes the map jarring in comparison. Otherwise, nicely done!

Author Response

Dear Reviewer,

The conceptual diagrams are adjusted and fully added elements such as source notes, data source, and author. 

I am grateful for your review.

Reviewer 3 Report

Simply put, I could not understand what the authors wanted to convey in this paper. There are too many sentences that are to understandable. This paper also needs more logical construction of argument. Its focus is not yet clear. At this poor condition, it is impossible to assess the value of this paper for publication. I strongly recommend the authors have editorial services before considering the submission of this paper.

Author Response

Dear Reviewer, 

Please kindly see my response as below:

Point 1: Simply put, I could not understand what the authors wanted to convey in this paper. There are too many sentences that are understandable.

Response 1:

Title of article: The author has re-research all the contents of the manuscript, which is considered a new form of the circular economy. Therefore, the title of the revised manuscript will be Circular Economy: A Study on the Use of Airbnb for Coastal Sustainable Development in the Vietnam Mekong Delta.

Abstract: The manuscript has been revised with research objectives, the aim, the tasks of the paper research in the abstract. Furthermore, instrumental tools and – what type of factor analysis and what type of multiple regression analysis are specified.

Literature: The author has added arguments about the circular economy in Vietnam's Mekong delta, the circular economy and tourism, sustainable development goals and their relevance to the topic, tourism sustainable development, its relationship with coastal resource protection, and sustainable consumption.

Discussion: The author provides some insights about the local government's plans to develop sustainable tourism practices and what are they? What resources are available for the traditional Airbnb business and the new model in the new form of the circular economy by describing in detail the tourism characteristics of some provinces in the Mekong Delta. Such performances have pointed out specific differences in the target audience of the new model, leading to a difference from traditional Airbnb.

Point 2: This paper also needs a more logical construction of the argument. Its focus is not yet clear. In this poor condition, it is impossible to assess the value of this paper for publication. I strongly recommend the authors have editorial services before considering the submission of this paper

Response 2: After the manuscript was adjusted, the author asked for the support of MDPI editorial services and received more positive feedback. Thanks for your comments and suggestions.

I am grateful for your review.

Reviewer 4 Report

Dear authors,

Thank you so much for the opportunity to review the manuscript entitled “Coastal Interconnection: A Study on the Use of Airbnb for Coastal Sustainable Development in the Vietnam Mekong Delta”.

The paper addresses an interesting issue of the relationship between sustainability and circular economy. The topic fits well with the current situation and seems to be suitable for the Sustainability editorial line.

However, I have concerns about the execution of this paper, as illustrated in details below. In any case, I think there are some areas where improvement can be made. Hope my suggestions are useful.

  1. In the Abstract: the authors have to specify what instrumental tools and what does the structural equation model includes – what type of factor analysis and what kind of multiple regression analysis? Furthermore, the research objectives, the aim, the tasks of the paper research are not clearly stated. State clearly the relation between the coastal resources, circular economy and sustainable development! I believe the authors should be more specific to the topic while conducting a literature review and discussing the results. Is there a demand for such type of accommodation in the delta? What is the ecological footprint of Vietnam Mekong Delta tourism – provide evidence?

In my opinion, the Introduction has to be separate from the Literature review.

  1. I believe the author should be more specific to the topic while conducting a literature review and discussing the results.

The model and the methodology need to be elaborated further, provide more scientifically proven research on the topic. What is more, the hypothesis formation should take place after the literature review. Which one is the mediator variable in the mediation analyses? Which of the hypothesis is accepted and which one rejected?

  1. The authors need to elaborate further on:

Circular economy with relation to the tourism sector; sustainable development goals, their relation to the topic; tourism sustainable development and its relation with coastal resources protection; Hypothesis development and elaboration; Amos 20; structural equation modelling; Mediation analyses?

  1. In the Discussion – please provide some insights is the government/local government plans to develop sustainable tourism practices and what are they? What resources are available for traditional Airbnb business and what do you mean by for traditional Airbnb business/ the innovated model and what is the difference between them?
  2. The Conclusion – what is the practical application of the research? Provide insights on how the findings can be helpful to all stakeholders?

Why the numbering of the lines starts from page 14?

Some statements must be more precise and therefore revised:

This study presents a model based on the case of using Airbnb in the Vietnam Mekong Delta, where there are abundant coastal ecosystems including aquaculture, agriculture, fishing, and seasonal fruits but these resources have not yet been fully taken into account to foster coastal development.

More specifically, the global poverty rate is significantly increased from 88 million to 115 million people into extreme poverty which defined on less than 1.90 USD as the disruption of the ongoing COVID-19 pandemic which known as the coronavirus pandemic outbreak from Vu Han, China at the end of 2019.

However, social exclusion (working from home) and policy of isolation (for tourism and import-export) in the first three months of 2020 as an extremely heavy economic pressure for coastal farmers have to confront

Nevertheless, some potential issues of sharing economy have also been researched on fear of strangers and social problems by interaction between local residents and tourists, following by sharping trust and belief in rural areas

lack the consideration of coastal interconnection in sustainable development to cope with human impact on the environment and steep decline

Literature on the sharing economy has been emphasized as a sustainable consumption, which is significantly supported by cost-saving features benefits in minimizing advertising costs and transaction costs based on the integration of technology performance and enterprise management [20,21]; therefore, as a technological intermediary, the decision to use Airbnb was based on elements such as sustainability, enjoyment, and economic benefits between those who want to rent out space and those who are looking for a space to rent [22].

As evidenced by a number of case studies and practical applications, Wu and Shen [26] illustrated that Airbnb - when stating that your argument is supported by a number of studies and practical applications, the Authors need to cite at least three sources or revise the statement.

The hosts on Airbnb must be open to meeting with unknown people from different countries; thus, trust has a vital role in the Airbnb platform because of sharing and the fact that a financial transaction is suggested by a third party, with technological assurances. Tussyadiah and Pesonen [30] posited that collaborative consumption means trusting strangers, and the lack – of what?

Thus, it is hypothesized that whether? local residents’ expected Airbnb impacts would support local sustainable development.

Grammar, spelling and lexical constructions have to be revised – neighboring; tourism industry

However, social exclusion (working from home) and policy of isolation (for tourism and import-export) in the first three months of 2020 as an extremely heavy economic pressure for coastal farmers have to confront

Literature on the sharing economy has been emphasized as a sustainable .. – the literature on a topic cannot BE emphasized; only one definition on sustainable consumption?

Hypothesis 5. And Hypothesis 6 have the same abbreviation – it is confusing

Author Response

Dear Reviewer,

Thank you for your comments.

Round 2

Reviewer 3 Report

A review of revised “The Circular Economy: A Study on the Use of Airbnb for Coastal Sustainable Development in the Vietnam Mekong Delta”

Overall comment

This paper appears to achieve too many things in one paper. The objective of this paper, if one reads from the beginning to the end, is not yet clear. Its interpretation/explanation about the results of the questionnaire survey needs much more clarification (for this they can read a number of good papers that discuss questionnaire results and learn from them). Compared to the multifaceted statistical methods it offers, the survey design and questions appear to be superficial and premature to understand the connection between Aribnb, ecotourism and a circular economy. As suggested to the previous submission, some technical problems remain, unfortunately. Many terminologies were loosely employed and some terms were used interchangeably without careful attention to the implications of the terms (see below). Added sentences contain so many typos and unclear sentences. It sounds premature to consider its publishability. Although it costs, I do strongly recommend that the authors invest in editing services and obtain additional peer opinions from colleagues to refine the design and overall flow of this paper before submitting to any journal. I made additional comments to some sections below.

Title

Not clear what it aims to do. Need to connect the circular economy and Airbnb.

Abstract

It says that the aim is to propose a coastal sustainable development structure model with the focus on a sharing economy and ecotourism. They added ideas of three modes: positive economic effects, positive economic pressures, coastal sustainability. Please ask if your conclusion and much of the main discussion about the results really respond to this aim.

Introduction

Except the last three paragraphs, the authors here discuss wide-variety of topics without much connection to the circular economy, ecotourism or Airbnb (so far nothing said about the importance of Airbnb for ecotourism). They need to restructure the introduction by rearranging information within a context of costal sustainable development structure, sharing economy or ecotourism. For example, how does the fishery discussion in the second paragraph help readers understand the importance and novelty of this paper. Each paragraph tends to discuss multiple topics without topic sentence. This rough writing alone is confusing and makes this paper inhibiting to understand its real value and contribution. At page 4, the authors mention that they want to incorporate the sharing economy concept into ecotourism. If this is what they want, the above discussion in the introduction should discuss why this incorporation is necessary and how this approach can provide a new insight to a specific scholarship. The following discussion at page 4 within the introduction section does show the authors’ attempt to emphasize the significance of their approach, but their explanation falls far short of achieving the purpose. Regarding the significance of this paper, the authors in the conclusion claim that this paper offers “the first attempt to analyze the integration of the sharing economy and local ecotourism.” I am not convinced yet. The introduction can help understand this point.

Another point is that much of the first half of the discussion in the introduction can be done in the subsection for “study area.”

Literature review

The authors tend to discuss multiple topics in one paragraph without clear topic sentence. Overall, this subheading, “literature review,” sounds misnomer as it tends to simply inform about such terms as a sharing economy, sustainability, resource sharing, integrated development, and Airbnb’s popularity (all in one paragraph!). Sort information and make topics connected much more closely to the objective and rationale.

This section also discusses hypotheses. This discussion sounds more appropriate to be done in methodology, not in literature review.

Added sentences in this section contain many careless typos. The authors should obtain an editorial service to clean these and make this paper readable at least.

Method

The authors claim that the respondents to their questionnaire survey “participated in the survey enthusiastically and responsibly.” Was this survey done in the midst of the COVID-19 pandemic in person? If not, how do the authors know the respondents were enthusiastic? Also, explain here how many were asked and how many persons responded. The authors say that 26 questions were asked. Then in the “Results” section (the first paragraph at 4.1), they say 303 questions were asked. Which is correct?

Results

The discussion of the results at page 12 needs clarification in many areas. So far it sounds to me that the authors are deliberately emphasizing the potential benefit of using Airbnb. For example, the second sentence at page 12 (“Relating to positive constructs, the survey report illustrated that coastal residents … had high expectations for … Airbnb.”) argues the importance of Airbnb, but the following sentences do not seem to buttress this point. If the authors argue otherwise, they need to explain what evidence made them think about “high expectation” among the respondents. In the next paragraph, the authors explain that “most local residents disagreed that this circular economy of the sharing economy [awkward expression here] would have a positive influence on their living conditions.” This sentence sounds contradictory to the argument in the previous paragraph. As far as I understood, the authors argue that Airbnb can be a tool to enhance a circular economy in the Mekong Delta region.

Another interpretation problem the authors have here on Airbnb was the last few sentences at page 12, where they argue that 35% and 30% of the respondents in the agriaquaculture sector did not think that Airbnb would not cause disturbance. Why do they now merely highlight this sector alone? Also, what happened to the rest of 45%? Also, I do not understand the salience of this question. What disturbance do the authors in mind to be caused by this internet tool? Much clearer explanation is needed.

Suspicious contents

Page 2, 2nd paragraph: Most part of the following sentence was copied from an internet source: https://www.vietnam-briefing.com/news/investment-environment-mekong-delta.html/ but not cited.

Page 2, 1st paragraph (6th line): “More specifically, there have been an additional 88 to 115 million people falling into extreme poverty….” Is this number within the coastal ecosystems? The World Bank document does not seem to be specifying the area. Sounds misleading here. Overall, this paragraph sounds not clear about the point the authors are making. Do they want to talk about the richness of coastal ecosystems? Do they want to emphasize poverty in coastal zones?

Author Response

Dear Reviewer, 

I would like to sincerely thank you for taking your valuable time to review and give me valuable comments.

This manuscript is a resubmission of an earlier submission. The following is a list of the peer review reports and author responses from that submission.